# Life Unknown: Preliminary Scheme for a Magnetotrophic Organism

**DOI:** 10.3390/life13071446

**Published:** 2023-06-26

**Authors:** Dirk Schulze-Makuch, Louis N. Irwin

**Affiliations:** 1Astrobiology Group, ZAA, Technische Universität Berlin, Hardenbergstr. 36A, 10623 Berlin, Germany; 2German Research Centre for Geosciences (GFZ), Section Geomicrobiology, 14473 Potsdam, Germany; 3Department of Plankton and Microbial Ecology, Leibniz Institute of Freshwater Ecology and Inland Fisheries, 16775 Stechlin, Germany; 4School of the Environment, Washington State University, Pullman, WA 99163, USA; 5Department of Biological Sciences, University of Texas at El Paso, El Paso, TX 79968, USA; lirwin@utep.edu

**Keywords:** magnetotroph, life as we do not know it, metabolism, extremophile

## Abstract

No magnetotrophic organism on Earth is known to use magnetic fields as an energy source or the storage of information. However, a broad diversity of life forms is sensitive to magnetic fields and employs them for orientation and navigation, among other purposes. If the magnetic field strength were much larger, such as that on planets around neutron stars or magnetars, metabolic energy could be obtained from these magnetic fields in principle. Here, we introduce three hypothetical models of magnetotrophic organisms that obtain energy via the Lorentz force. Even if an organism uses magnetic fields only as an energy source, but otherwise is relying on biochemistry, this organism would be by definition a magnetotrophic form of life as we do not know it.

## 1. Introduction

No organism on Earth that uses magnetic fields as an energy source or an information storage and transmission mechanism is known. The reason for this is likely related to the general weakness of Earth´s magnetic field compared to other available energy sources, such as chemical and light energy. We have previously compared the amount of energy that could be gained from a magnetic field with the amount extractable from light and chemical sources [1,2]. We calculated that the Earth’s magnetic field could produce 3 *×* 10^−11^ eV per unit charge from the Lorentz force, which is about 11 orders of magnitude lower than the energy that can be obtained from a chemoautotrophic reaction or a photon on Earth.

However, there are stellar bodies with much stronger magnetic fields such as neutron stars or magnetars. The magnetar SGR 1806-20, for example, has been reported to have a magnetic field strength of about 10^15^ Gauss, or 10^11^ T [3], which is nearly sixteen orders of magnitude larger than Earth´s magnetic field. Whether any life could originate and persist in such strong magnetic fields is unknown and, of course, highly speculative. However, for organisms originating and evolving under those conditions, magnetic fields of such a strength would have profound effects on biology and the functioning of a cell [4]. The quantity of energy available on an exoplanet in orbit around a magnetar such as SGR 1806-20 would likely be favored over both chemical and light energy, especially if those sources of energy are limited.

The purpose of this paper is to flush out the possibility with some concrete examples of how biomagnetism could be harvested for energy production.

## 2. Known Organismic Uses of Magnetic Fields

Although magnetic fields on Earth are not used by organisms for energy gain or genetic storage and replication, life on our planet is sensitive to magnetic fields. This includes animals, plants, and even microbes. Various animals, as different as birds and lobsters, use magnetic positional information to stay on track along migratory pathways and take up food at specific locations [5]. This finding is further substantiated by magnetic anomalies, which are usually caused by near-surface geological features which disrupt, for example, the navigational cues for the homing behavior of pigeons [6,7]. The homing behavior to natal areas in sea turtles, salmon, and some birds has been shown to be the product of an imprint on the magnetic field of the natal location in the juvenile organism, which facilitates the organism’s return as an adult [4]. Plants are sensitive and responsive to varying magnetic fields by altering their gene expression and phenotype [8]. The most receptive microbes to magnetic fields are magnetotactic bacteria. These bacteria produce magnetosomes that enable orientation and migration along magnetic field lines toward preferred oxygen concentrations [9]. The magnetosomes contain iron-rich magnetic particles within a lipid bilayer membrane, which have high chemical purity and near-perfect crystalline structures, narrow size ranges, species-specific crystal morphologies, and specific arrangements within the cell (mostly as chains). Chains of magnetite crystals which could represent magnetosomes have also been found in the Martian meteorite ALH84001 [10], which could indicate bacteria on Mars that oriented themselves to an early magnetic field on Mars [11]. Either way, the sensitivity of life to magnetic fields seems to be a common property of life forms.

## 3. Use of the Lorentz Force Magnetic Fields

While no organism on our planet is known to extract energy from magnetic fields, either as a main or supplementary source of energy, in principle, this is possible. Free energy can be extracted from magnetic fields via charge separation, for example, via the Lorentz force or via induction [1,2]. If a model organism of a magnetotroph is exposed to the Lorentz force, that force can be expressed as
F_L_ = q(E + v × B)(1)
where E is the electric field acting on the charge (Newton/Coulomb; N/C), v is the velocity (m/s) of the charge in the magnetic field, and B (T) is the magnetic field strength. The cross product v *×* B is reduced to vB in the special case of a perpendicular movement of the charge with respect to the direction of the magnetic field B. If the movement of the charge occurs parallel to the direction of the magnetic field, the cross product is zero, and thus, in the absence of an electric field, no force acts on the charge q. In the absence of a magnetic field (B = 0), a charge is accelerated parallel to the electric field such that
F = q E(2)
with E being the magnitude of the electrical field (N/C).

It should be noted that Equations (1) and (2) apply to the force from a single charge, whereas a living cell would involve the activity of millions of charges acting in parallel. Also, rather than thinking about a cell in micrometer dimensions, the energy yield could be increased manyfold or even by orders of magnitude if the organism is elongated to lengths in the millimeter or centimeter range.

The ability of magnetic energy to affect the movement of charged particles is well known. Contrary to electric forces that drive cellular machinery, the magnetic forces available in our solar system are too small to directly affect cell functions [12]. However, if cells are experimentally exposed to magnetic fields as strong as a few Teslas (T), many biological effects can be observed. Zablotskii et al. [3] predicted that if the magnetic field strength is increased to 100–1000 T, the cell membrane potential can be changed by a few mV, and the magnetically assisted intracellular diffusiophoresis of large proteins can occur. It has been known for some time that large uniform magnetic fields can affect the diffusion of charged ions (e.g., Na^+^, K^+^, and Ca^++^) [13,14] and polarize particles, which can increase their clustering via dipole interactions [15]. It has also been shown that a large uniform magnetic field can stimulate the diffusion of paramagnetic and diamagnetic species inside cells [3], leading to significant concentration gradients [16]. Magnetic fields with large spatial gradients can impose enough force on paramagnetic and diamagnetic particles to alter cellular functions [17]. In particular, paramagnetic elements such as O, Al, and Ti can become polar and be affected by a strong enough magnetic field. This is the case for most chemical elements and compounds that have unpaired electrons. Even diamagnetic elements and compounds, which do not have unpaired electrons, are moved to the area of the lowest magnetic field strength if subjected to a very strong magnetic field gradient [18].

Electromagnetic induction generates energy through a periodically changing magnetic field. The Lorentz force, however, still applies, as a charge can move perpendicular to the induced magnetic field, but the energy yield is much lower in magnitude. For example, some energy is generated in a subsurface salty ocean like that of Jupiter’s moon Europa due to the oscillatory magnetic field of Jupiter.

However, while induction is an intriguing consideration for very specialized environmental scenarios such as the Jovian system [1], the energy that can be obtained by the Lorentz force is under most scenarios many orders of magnitude larger than by induction. Thus, in the following, we focus on the Lorentz force as a means for harvesting free energy in our hypothetical organisms.

## 4. Hypothetical Ways in Which a Magnetotroph Could Work

The following examples of hypothetical organisms show how magnetic force lines could be harvested for the synthesis of high-energy biomolecules that fuel living processes. All three examples depend on the Lorentz force imposed upon charged particles in a magnetic field, and all three rely on the tendency for living cells, whether freely rotatable or growing on a stable substrate, to become oriented with respect to a magnetic field [19,20].

### 4.1. Magnetotroph Model #1

Assume an ellipsoid organism is divided into two compartments by a midline membrane permeable to cations and anions through ion-specific channels (Figure 1). Further assume that the organism is free-floating in a medium such as an aqueous environment that erratically causes the organism to tumble through cycles of rotation through both perpendicular and parallel orientations relative to the magnetic force field. When momentarily oriented with the organism’s long axis perpendicular to the magnetic force field, cations will be driven to the right and anions will be drawn to the left (Figure 1a). This leads to an excess of cations in the right compartment and anions in the left compartment.

As the organism rotates toward parallel alignment with the magnetic force field, the magnetic force decreases, and diffusion potentials from concentration gradients rise (Figure 1b). At some point, those two forces balance, resulting in no net flux of ions through the membrane.

As the organism’s alignment becomes parallel with the magnetic force field, the magnetic force drops to zero, and the diffusion potential from concentration gradients causes ions to flow down their concentration gradients unimpeded (Figure 1c). As each ion moves through its selective channel, the exothermic energy of diffusion is coupled to endothermic molecular machinery that forms a high-energy bond, such as the phosphorylation of a protein (X*~P*). Coupling the energy yield from ionic diffusion down concentration gradients to form high-energy chemical bonds is a common biochemical process in organisms on Earth. ATP synthesis from proton diffusion in chemiosmosis is a prototypical example.

As the organism rotates back toward perpendicular alignment with the magnetic force field, the magnetic force increases and the diffusion potential from concentration gradients falls. At some point, these two forces balance, resulting in no further net flux of ions through the membrane (Figure 1d). When the organism has oriented its long axis perpendicular to the magnetic force field again, the cycle will repeat with cations driven to the right and anions drawn to the left (Figure 1a). The periodicity of the rotation is irrelevant so long as the organism becomes oriented occasionally in a manner that generates asymmetric concentration gradients across the internal membrane.

A potential complication of this scheme could be a consequence of Lenz’s Law, which predicts that a change in voltage induced by the electrical current of the moving ions opposes the direction of the Lorentz force. We assume that changes in voltage induced by the flow of ions over microscopic distances would be negligible in comparison to the powerful electromagnetic field driving oppositely charged ions in opposite directions by the Lorentz force.

It should be noted that ionic currents are prominent features of biological organisms as we know them. They are involved in many aspects of cellular function, including inter- and intracellular signaling, the establishment and dynamic properties of excitable membranes such as (but not exclusively) those of nerve cells, the homeostatic balance of bodily fluid composition and osmolarity, nearly all sensory mechanisms, muscle contractions and therefore movement, and the energy transformations of intermediary metabolism, as envisioned in this and the next model organism.

### 4.2. Magnetotroph Model #2

Assume oval-shaped organisms are living on a ferromagnetic substrate with the abundant availability of molecular sulfur, iron compounds, and water (Figure 2a). The organisms orient with ion-selective channels facing toward or away from (perpendicular to) a strong magnetic field (Figure 2b). Molecular sulfur, ferric ions (Fe^3+^) from iron-bearing minerals, and water are engulfed, ingested, or absorbed from the substrate into the organism. A role of the Lorentz force propulsion of charged particles such as ferric ions could also be to promote influx. An exothermic reaction occurs in which sulfur is oxidized and ferric ions are reduced, yielding energy that can be used for various metabolic, biosynthetic, and kinetic processes (Figure 2c). Negatively charged bisulfate ions are drawn through selective anionic channels, and ferrous (but not ferric) ions are pushed through selective cationic channels by the magnetic force field. This movement of ions through their respective selective channels engages molecular machinery that adds a high-energy phosphate bond to yield an energy-bearing molecule (Figure 2d), adding to the bioenergetic inventory along with the energy resulting from the redox reaction that generated the ions in Figure 2c.

### 4.3. Magnetotroph Model #3

Assume a long, slender cellular microbe is lined with rows of cilia-like structures on either side (Figure 3a). While many different morphologies could be imagined, a slender cellular structure has the advantage of being able to link opposite sides of the organism through a long row of parallel microscopic molecular strands. As in model #1, this organism is assumed to tumble erratically through cycles of parallel and perpendicular alignment to the magnetic force field. The cilia on one side contain positive charges at their tips, while those on the other side carry negatively charged tips. A band of elastic molecules connects with molecules at the base of the cilia on either side of the cell. The intercellular space contains molecules (X-Z) with low-energy bonds when the microbe is lined up in parallel with the magnetic force field (Figure 3a).

When the microbe reorients perpendicular to the magnetic field, the magnetic force bends one row of cilia forward and the other row of cilia backwards. Gear-like molecular machinery in the membranes causes conformational changes that pull the connecting elastic bands in opposite directions at either end, causing the bands to stretch (Figure 3b).

When the organism resumes an orientation parallel to the magnetic field, the elastic bands recoil to their unstretched conformation. The tensile energy thus released is coupled to the formation of high-energy bonds in energy-rich carrier molecules (X~Z) within the intercellular space (Figure 3c).

The high-energy compounds thus created (X*~Z*) can then be used to drive other endergonic metabolic and biosynthetic processes (A→B, C→D, etc.) returning the energy carriers to their low-energy state (X-Z, Figure 3d).

## 5. Discussion and Conclusions

Given the sensitivity of life on Earth to magnetic fields, it seems plausible that biological systems could make use of magnetic fields if exposed over evolutionary time to higher field strengths than those experienced on Earth. What adaptations to this type of environment would be possible and which would not is unknown, but the harvest of free energy, as shown in our three magnetotrophic models, seems theoretically plausible. What we have sought to demonstrate is that the use of magnetic-derived energy by an organism seems feasible in principle if the magnetic field is strong enough.

The use of magnetic fields for reproduction seems even more speculative, but the scheme by Feinberg and Shapiro [21] provides an intriguing theoretical example of how life could work with a magnetic rather than a chemical code. They suggested a coding scheme based on the alignment of magnetic particles, which could store information and under certain circumstances also replicate it.

The influence of magnetic fields on charge separation across semipermeable membranes is constrained by the lack of magnetic field influence on elements or chemical compounds that do not have unpaired electrons. However, even if the organism would use magnetic fields only for energy gain but otherwise would employ conventional chemical reactions for its other metabolic processes, we would still call it a magnetotrophic organism, by analogy with a phototrophic organism that uses light as its energy source but relies otherwise on biochemistry for its other functions. Also, if such an organism did exist, it would still count as a form of life, since it would exhibit the basic characteristics of life as traditionally defined (Table 1), as well as satisfy our previously proposed definitions of life [1,22,23].

## Author Contributions

Conceptualization, D.S.-M.; models of hypothetical organisms L.N.I.; writing—original draft preparation, D.S.-M.; writing—review and editing, D.S.-M. and L.N.I.; schematic visualization, L.N.I. All authors have read and agreed to the published version of the manuscript.

## Figures and Tables

**Figure 1 life-13-01446-f001:**
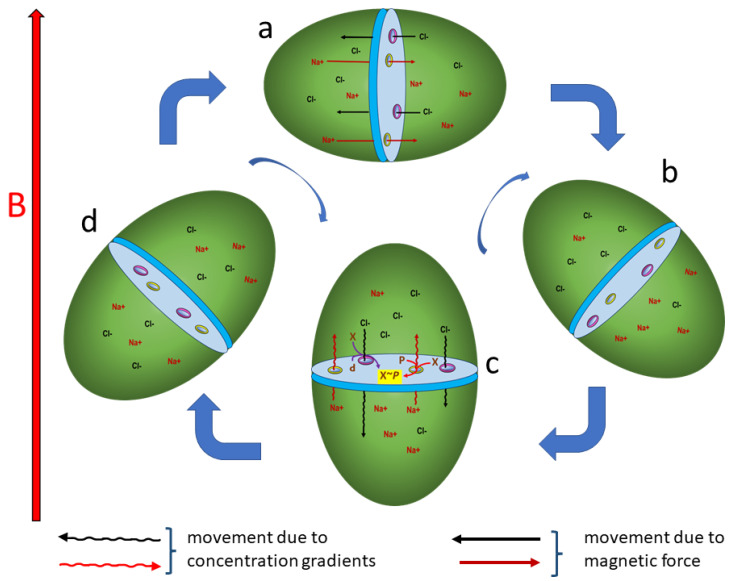
Magnetotroph #1. (**a**) Ions flow through selective channels in opposite directions when organism is oriented perpendicular to a magnetic field B. (**b**) Ionic fluxes cease when organism aligns at 45° angle to magnetic field. (**c**) Lack of magnetic force when organism aligns parallel to force lines allows ionic diffusion down concentration gradients coupled to phosphorylation, yielding high-energy biomolecules. (**d**) Ionic diffusion is balanced by returning magnetic force when organism passes through 45° angle to magnetic field.

**Figure 2 life-13-01446-f002:**
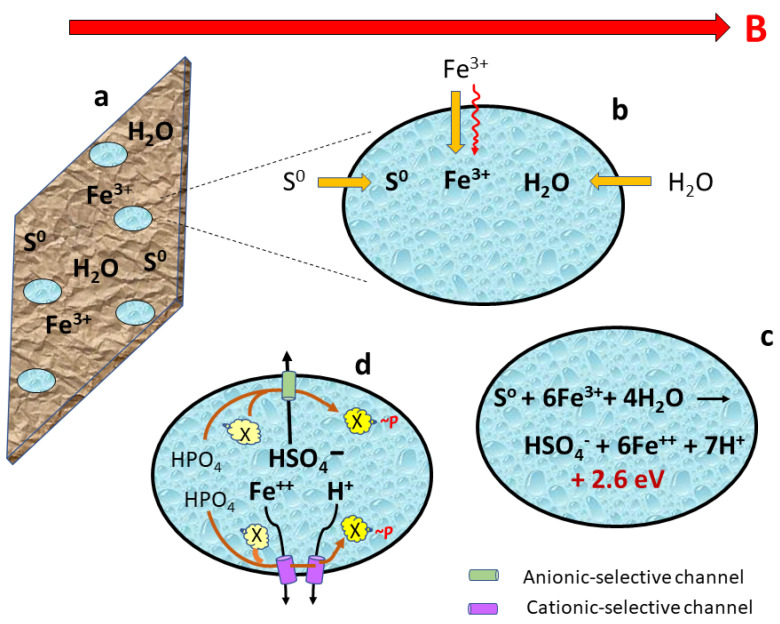
Magnetotroph #2. (**a**) Flat oval organisms dwelling on a ferromagnetic substrate align parallel to a magnetic force B. (**b**) Sulfur, iron, and water are ingested or seep into organism, with assistance from magnetic field for ferric ions. (**c**) Reduction of ferric to ferrous ions and oxidation of sulfur generates new ions and releases energy. (**d**) Bisulfate and ferrous ions are driven in opposite directions by magnetic forces through selective channels. This ionic flux is coupled to molecular machinery that generates more energy storage in the form of high-energy biomolecules (X~*P*).

**Figure 3 life-13-01446-f003:**
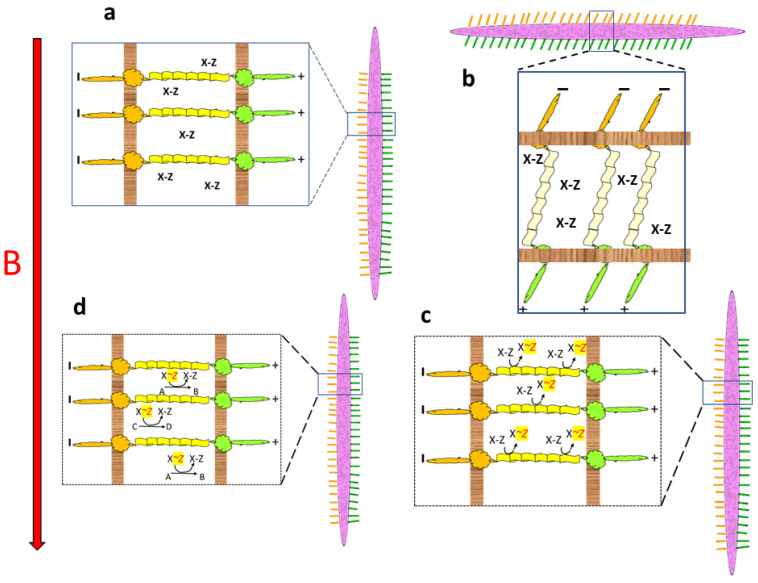
Magnetotroph #3. (**a**) A long slender cellular microbe (purple) lined with rows of cilia-like structures tipped with positive charges on one side and negative charges on the other. The cilia are connected through elastic molecular strands surrounded by low-energy biomolecules (X-Z), as shown in expanded view. (**b**) When the microbe aligns perpendicular to the magnetic field B, cilia are bent in opposite directions on either side of the microbe, and a gear-like mechanism at the base of the cilia causes molecular strands connecting the cilia from opposite sides to stretch. This provides binding sites for X-Z molecules. (**c**) When the microbe returns to a parallel alignment with the magnetic field, the elastic strands passively recoil, and the released tensile energy is coupled to a phosphorylation reaction that creates a high-energy biomolecule (X~*P*). (**d**) The high-energy X~*P* biomolecules can then catalyze endergonic reactions (A→B, C→D) required for other living processes.

**Table 1 life-13-01446-t001:** Properties of biological organisms in life as we know it compared to a magnetotrophic analog.

Property	Basic Requirement	Life as We Know It	Magnetotroph
Metabolism	Enegy obtained from electron transfer	Various biochemical pathways including photosynthesis; energy storage as high-energy phosphates	Energy uptake using charge separation and re-combination via Lorentz force or by induction
Growth	Increase in size of single unit	Cell growth as long as nutrients are available and environmental conditions are favorable until reproduction occurs. Self-organizing as development proceeds; errors corrected by enzymes	Growth of biomolecules sensitive to magnetic field and/or magnetite crystals as long as favorable environmental conditions prevail and a sufficient ion source is present
Reproduction	Storage and replication of information	Genetic information stored in chemical form and replicated metabolically	Genetic information stored in alignment of magnetized components and replicated through parallel alignment of new magnetized components [21]
Adaptation to Environment	Compensation for, reaction to, and development of new abilities in response to environmental changes	Genetic adaptation over time through mutations, transposition of genetic material, transformation, conjugation, and transduction. Homeostatic adaptation within individuals: movement to nutrient source, cellular shrinkage or formation of spores in nutrient-poor environment, and enzyme induction	Adaptation through genetic toolset and adjustments to chemical and physical changes in the external environment including characteristics of the magnetic field

## Data Availability

No additional data other than that in the paper.

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
