# Peer review of "Life Unknown: Preliminary Scheme for a Magnetotrophic Organism"

_life, 2023, doi:10.3390/life13071446_

Round 1

Reviewer 1 Report

The manuscript "Life unknown: Preliminary scheme for a magnetotrophic organism" is an interesting proposal about how an organism can use strong environmental magnetic fields as energy source. I consider this paper as an valid ideia and that it deserves to be published in Life journal.

I have some minor comments to the authors about the text;

1.line 47. 10^15 Gauss = 10^11 Tesla

2. line 47. Maintain the reference style using number. The reference Ibrahim et al. is not present in the reference list.

3. line 48. The geomagnetic field is about 16 orders of magnitude lower with respect to 10^15 Gauss

4. line 69... [4])...remove )

5. line 121. "However, compounds that do not..." Here you mean that only paramagnetic molecules must feel the effect of a strong magnetic field. However, very strong magnetic fields affects all matter through diamagnetism. I believe you can rewrite better that phrase considering the diamagnetism of the molecules. Remember: all matter is diamagnetic!

 6. line 143.... {19,20]...change { by [

7. line 145, model #1. Describing the model, you only ask that the long axis of the cylindrical body be perpendicular to the magnetic field. But the Lorentz force includes the velocity. If ions are concentrating to the right and to the left of the magnetic field is because their velocity is perpendicular to the magnetic field. How you guarantee that condition? If you left free, ions also can concentrate in the celular walls located in the perpendicular direction to the magnetic field, i.e., in the direction out of the plane where the cell is drawn.

Line 181 and Figure 2(a)... the organism drawn looks parallel, not perpendicular, to the magnetic field

Lines 185, 187, 192 and 193: change (b), (c) and (d) by (Fig. 2b), (Fig. 2c) and (Fig. 2d)

Author Response

The manuscript "Life unknown: Preliminary scheme for a magnetotrophic organism" is an interesting proposal about how an organism can use strong environmental magnetic fields as energy source. I consider this paper as an valid ideia and that it deserves to be published in Life journal.

I have some minor comments to the authors about the text;

Thanks so much for the constructive review !

1.line 47. 10^15 Gauss = 10^11 Tesla Done

  1. line 47. Maintain the reference style using number. The reference Ibrahim et al. is not present in the reference list. The correct reference has been numbered in the text and inserted into literature cited.

  1. line 48. The geomagnetic field is about 16 orders of magnitude lower with respect to 10^15 Gauss corrected

  1. line 69... [4])...remove ) Done

  1. line 121. "However, compounds that do not..." Here you mean that only paramagnetic molecules must feel the effect of a strong magnetic field. However, very strong magnetic fields affects all matter through diamagnetism. I believe you can rewrite better that phrase considering the diamagnetism of the molecules. Remember: all matter is diamagnetic!

Thank you – the text has been rephrased

  1. line 143.... {19,20]...change { by [ Done

  1. line 145, model #1. Describing the model, you only ask that the long axis of the cylindrical body be perpendicular to the magnetic field. But the Lorentz force includes the velocity. If ions are concentrating to the right and to the left of the magnetic field is because their velocity is perpendicular to the magnetic field. How you guarantee that condition? If you left free, ions also can concentrate in the celular walls located in the perpendicular direction to the magnetic field, i.e., in the direction out of the plane where the cell is drawn.

It is correct that not all free ions will cross the semipermeable membrane under the influence of the magnetic field, but all that the model requires is that more cations concentrate on the far side of the membrane and that more anions concentrate on the near side of the membrane to some degree. This sets up concentration gradients of potential energy that can be harvested to form high-energy chemical bonds when the ions spontaneously flow down their concentration gradients as the organism aligns parallel to the magnetic field (neutralizing the Lorentz force).

Line 181 and Figure 2(a)... the organism drawn looks parallel, not perpendicular, to the magnetic field.

Correct, but the key point is that the ion-selective channels are oriented such that the trajectory of ion movement is perpendicular to the magnetic field. The text has been reworded to clarify this point.

Lines 185, 187, 192 and 193: change (b), (c) and (d) by (Fig. 2b), (Fig. 2c) and (Fig. 2d) 

Done. Thanks for catching this.

Reviewer 2 Report

Life unknown: Preliminary scheme for a magnetotrophic organism

Dirk Schulze-Makuch and Louis N. Irwin

            The title is intriguing but the text doesn’t deliver on the promise.  The lead in: Introduction, Known Organismic Uses of Magnetic Fields and Use of the Lorentz Force Magnetic Fields are a good start however the hypothetical organisms are a disappointment.

Magnetotroph Model #1- Describes a cylindrical organism rotating in a ‘strong’ magnetic field.  Why the shape of the organism (cylindrical in text, ellipsoid in Fig. 1) matters is not clear.  There is no indication of why the organism rotates or the regularity of its rotation.  It is this rotation that generates the energy, the magnetic field is merely the background.  Without an explanation for the rotation there is no explanation for energy produced by the magnetic field.  Contrary to the text the diffusion potential from the concentration gradient does not vary with rotation.  The mechanism coupling ion flow to phosphorylation, yielding high-energy biomolecules is not explained .

Magnetotroph Model #2- Describes organisms living on a ferromagnetic substrate.  Apparently, this model trades on the movement of ferrous ions in the magnetic force field.  This seems to simply provide some ion flow not a major component of energy flow.  The primary energy source is phosphorylation.  Again, the mechanism coupling ion flow to phosphorylation, yielding high-energy biomolecules is not explained.

Magnetotroph Model #3- This imaginative (bizarre) long slender cellular microbe lined with rows of cilia-like structures on either side is one of a myriad unusual structures that might be considered.  No explanation is offered as to the merit of this particular configuration.  The linkage of oppositely charged cilia-like structures on opposing sides used to produce high-energy biomolecule is interesting, but tangential to external magnetic field.  Like Model #1, it is the rotation that generates the energy, the magnetic field is the background and there is no indication of why or how the organism rotates.

Hypothetical mechanism for storage and replication of genetic information in a magnetotroph- Describes a purported mechanism for transmission of genetic information.  This mechanism has nothing to do with a high magnetic field.  In fact, the mechanism is utterly untenable.  As noted the stability of a chain of randomly oriented magnetic particles is utterly insufficient to act as a repository of genetic information.  Further the copying mechanism is untenable.  The force between the magnetic particles in the parent chain is equivalent to the force between the newly added progeny magnetic particles.  Thus, if a ‘newly replicated chain’ (duplex) were to be formed as shown in Fig. 4 (lower) there is no reason to expect that the parent and progeny chains could be cleanly separated.  With all the bonds of similar strength such a duplex would be expected to fragment rather than resolve into parent and progeny chains. 

A major concern that goes unaddressed in this manuscript is the short longevity of magnetars.  Their strong magnetic fields decay after about 10,000 years.  This is a remarkably short time during which to accomplish any meaningful evolution.

The English is fine.

Author Response

The title is intriguing but the text doesn’t deliver on the promise.  The lead in: Introduction, Known Organismic Uses of Magnetic Fields and Use of the Lorentz Force Magnetic Fields are a good start however the hypothetical organisms are a disappointment.

Magnetotroph Model #1- Describes a cylindrical organism rotating in a ‘strong’ magnetic field.  Why the shape of the organism (cylindrical in text, ellipsoid in Fig. 1) matters is not clear.

The shape of the organism doesn’t really matter (‘Cylindical’ is replaced by ‘ellipsoid’ in the text).  What does matter is the orientation of the ionic concentration gradients relative to the magnetic force field.

  There is no indication of why the organism rotates or the regularity of its rotation. It is this rotation that generates the energy, the magnetic field is merely the background.  Without an explanation for the rotation there is no explanation for energy produced by the magnetic field. 

Wording has been added to clarify that rotation is passive due to the motion of the medium which the organism inhabits. Energy is derived by resolving the diffusion gradients when the magnetic force field that created the gradients goes to zero.

Contrary to the text the diffusion potential from the concentration gradient does not vary with rotation. 

The concentration gradient increases when the long axis of the organism is perpendicular to the magnetic force field, and decreases when it parallels the force field. In other words, the concentration gradient does vary with rotation.

 The mechanism coupling ion flow to phosphorylation, yielding high-energy biomolecules is not explained. Any proposed mechanism in an alien organism evolved by an unknown evolutionary trajectory would be purely speculative and is beyond the scope of this paper, but mechanisms that harvest chemical energy from ionic gradients are well known in organisms on Earth.  Wording to this effect has been added to the text.

 .

Magnetotroph Model #2- Describes organisms living on a ferromagnetic substrate.  Apparently, this model trades on the movement of ferrous ions in the magnetic force field.  This seems to simply provide some ion flow not a major component of energy flow.

That argument could be made, since the oxidation of S and reduction of Fe+3 yield energy independent of the diffusion-driven phosphorylation mechanism.  Since this is only a model, there is no a priori justification for deciding whether diffusion-driven phosphorylation or the redox reaction independent of magnetic field influence provides the major component of energy flow. That some metabolic reactions contribute to energy flow independent of magnetic field is a reasonable assumption, and this model incorporates that fact, as the last sentence in Section 4.2 acknowledges.

The primary energy source is phosphorylation.  Again, the mechanism coupling ion flow to phosphorylation, yielding high-energy biomolecules is not explained.

See response  two paragraphs above, and text with additional new wording at the end of the third paragraph in Section 4.1.

Magnetotroph Model #3- This imaginative (bizarre) long slender cellular microbe lined with rows of cilia-like structures on either side is one of a myriad unusual structures that might be considered.  No explanation is offered as to the merit of this particular configuration. 

A myriad of morphologies might well be imagined. A new second sentence in Section 4.3 has been added to explain the virtue of a long slender shape.

The linkage of oppositely charged cilia-like structures on opposing sides used to produce high-energy biomolecule is interesting, but tangential to external magnetic field.  Like Model #1, it is the rotation that generates the energy, the magnetic field is the background and there is no indication of why or how the organism rotates.

As in model #1, the organism rotates passively due to motion of the medium which it inhabits. This is stated in a new third sentence in Section 4.3. The medium provides the force that causes the rotation, but this contributes no net gain in chemical energy for the organism. The energy gain comes from a mechanism tied to the magnetic force field.

Hypothetical mechanism for storage and replication of genetic information in a magnetotroph- Describes a purported mechanism for transmission of genetic information.  This mechanism has nothing to do with a high magnetic field.  In fact, the mechanism is utterly untenable.  As noted the stability of a chain of randomly oriented magnetic particles is utterly insufficient to act as a repository of genetic information.  Further the copying mechanism is untenable.  The force between the magnetic particles in the parent chain is equivalent to the force between the newly added progeny magnetic particles.  Thus, if a ‘newly replicated chain’ (duplex) were to be formed as shown in Fig. 4 (lower) there is no reason to expect that the parent and progeny chains could be cleanly separated.  With all the bonds of similar strength such a duplex would be expected to fragment rather than resolve into parent and progeny chains. 

We agree that there are a number of problems, including the ones pointed out here. Thus, we deleted the Feinberg and Shapiro model as a separate section of the paper. However, we mentioned it still briefly with a few sentences in the Discussion section because it remained until today the only model/scheme suggesting a reproduction mechanism not based on chemistry. Further, the Feinberg and Shapiro acknowledge the difficulties when they say “If the magnets placed along the chain can retain their alignment and be protected from the re-magnetization by an exterior field”  and we included now wording (in the figure caption explaining the scheme) that this would be indeed very challenging. At least in respect to reproduction (as pointed out above), not so much regarding storage, because magnetotactic bacteria have magnetic chains within their bodies and don´t have any difficulties to keep them aligned (though not as envisioned by Feinberg and Shapiro).  

A major concern that goes unaddressed in this manuscript is the short longevity of magnetars.  Their strong magnetic fields decay after about 10,000 years.  This is a remarkably short time during which to accomplish any meaningful evolution.

This is a valid concern, though it should be noted that we have only one example of an evolutionary trajectory, so don’t really know how long evolution might take under different planetary circumstances.

Reviewer 3 Report

Thank you for submitting the intriguing article exploring potential models for utilizing high magnetic fields in bioenergetics and genetic information storage and transfer. I appreciate the thought-provoking propositions presented. I would like to address a few points to further enhance the completeness of the work.

1. Regarding Model #1, which suggests a mechanism for maintaining a charge imbalance across a membrane, it would be beneficial to discuss the biological implications of such ionic current within an organism.

2. Additionally, it is important to explore the potential impact of Lenz's law-based currents that may arise during such rotation, as it could complicate the situation. I encourage revisiting this section and incorporating a discussion of these possibilities.

3. For Model #2, which focuses on magnetic field-assisted ion transport through membranes, it is crucial to consider the implications of orientation, especially in the context of any motion relative to the substrate. This raises questions about how organisms would navigate and exchange ions with the surrounding medium, particularly if they exhibit swimming behavior. Addressing these aspects would contribute to a more comprehensive understanding of the model.

4. It would be fascinating to hear the authors' perspective on the circumstances and challenges associated with multicellularity in such a high magnetic field environment. Exploring the potential adaptations and strategies employed by multicellular organisms in this context would provide valuable insights.

5. In living organisms, genetic material exists in the form of chromatin, which resembles a folded polymer. It would be insightful to discuss how such folding would impact the proposed Ising chain model for genetic material in this work.

Considering the complexities of chromatin structure and its influence on genetic processes within a high magnetic field environment would enhance the discussion.

Overall, I commend the authors on their work and appreciate the opportunity to provide feedback. I believe addressing these points will contribute to a more comprehensive and robust exploration of the potential models proposed in the article.

Author Response

Thank you for submitting the intriguing article exploring potential models for utilizing high magnetic fields in bioenergetics and genetic information storage and transfer. I appreciate the thought-provoking propositions presented. I would like to address a few points to further enhance the completeness of the work.

Thank you !

  1. Regarding Model #1, which suggests a mechanism for maintaining a charge imbalance across a membrane, it would be beneficial to discuss the biological implications of such ionic current within an organism.

A short paragraph has been added to this effect at the end of Section 4.1.

  1. Additionally, it is important to explore the potential impact of Lenz's law-based currents that may arise during such rotation, as it could complicate the situation. I encourage revisiting this section and incorporating a discussion of these possibilities. A short paragraph addressing this point has been added before the final paragraph in Section 4.1.

  1. For Model #2, which focuses on magnetic field-assisted ion transport through membranes, it is crucial to consider the implications of orientation, especially in the context of any motion relative to the substrate. This raises questions about how organisms would navigate and exchange ions with the surrounding medium, particularly if they exhibit swimming behavior. Addressing these aspects would contribute to a more comprehensive understanding of the model.

We envision that the organisms in model #2 would grow on a stable substrate in a direction that would always have them oriented so that their ion channels remain perpendicular to the magnetic force field. Note that no rotation is shown in Fig. 2. Therefore, as new diffusible ionic species are generated by intermediary metabolism, they will aways be forced out through their ion-selective channels by the Lorentz force from the electromagnetic field. The directional orientation of organisms in response to environmental factors is common, as illustrated by phototropism.

  1. It would be fascinating to hear the authors' perspective on the circumstances and challenges associated with multicellularity in such a high magnetic field environment. Exploring the potential adaptations and strategies employed by multicellular organisms in this context would provide valuable insights.

That is indeed an interesting question. In this paper, however, we have focused on how energy could be converted from electromagnetism to a usable chemical form, and that is at base a unicellular phenomenon. The physiology of multicellular organisms in high magnetic fields is another topic for another time.

  1. In living organisms, genetic material exists in the form of chromatin, which resembles a folded polymer. It would be insightful to discuss how such folding would impact the proposed Ising chain model for genetic material in this work. Considering the complexities of chromatin structure and its influence on genetic processes within a high magnetic field environment would enhance the discussion. We agree that this is an issue that raises problems we are not prepared to address within the scope of this paper, therefore we have decided not to retain this highly speculative idea in our revised version.

Overall, I commend the authors on their work and appreciate the opportunity to provide feedback. I believe addressing these points will contribute to a more comprehensive and robust exploration of the potential models proposed in the article.

Thank you so much for your constructive comments !

Round 2

Reviewer 2 Report

The deletion of the section 'Hypothetical mechanism for storage and replication of genetic information in a magnetotroph' is appropriate, however the retention of Fig. 4 defeats this remedy.

The English text is fine.

Author Response

Based on the reviewer´s comment, Fig. 4 has been deleted from the paper.